# Stress Superposition Method and Mechanical Properties Analysis of Regular Polygon Membranes

**DOI:** 10.3390/ma15010192

**Published:** 2021-12-28

**Authors:** Tao Peng, Qiuhong Lin, Bingyan Li, Ani Luo, Qiang Cong, Rongqiang Liu

**Affiliations:** 1Beijing Institute of Spacecraft System Engineering, Beijing 100094, China; tao11102021@163.com (T.P.); libingyan3@hotmail.com (B.L.); tao20211110@163.com (Q.C.); 2College of Mechanical and Electrical Engineering, Harbin Engineering University, Harbin 150001, China; luoani@hrbeu.edu.cn; 3State Key Laboratory of Robotics and System, Harbin Institute of Technology, Harbin 150001, China; iurq@hit.edu.cn

**Keywords:** stress superposition, membrane, curved edge, wrinkles

## Abstract

In this paper, the stress superposition method (SSM) is proposed to solve the stress distribution of regular polygon membranes. The stress-solving coefficient and the calculation formula of arbitrary point stress of regular polygon membrane are derived. The accuracy of the SSM for calculating stresses in regular polygonal membranes is verified by comparing the calculation results of the SSM with the finite element simulation results. This article is the first to propose a method to investigate the response of the arch height of the membrane curved edge to the membrane’s mechanical properties while keeping the effective area constant. It is found that the equivalent stress and the second principal stress at the midpoint of the membrane curved edge are effectively increased with the increase of the arch height of the curved edge. The second principal stress at the edge region of the membrane is relatively small, leading to the occurrence of wrinkles. When the stress at the midpoint of the curved edge is equal to that at the center of the membrane, the membrane plane attains the maximum stiffness and reduces the possibility of wrinkling at the edge.

## 1. Introduction

With the current rapid development of the aerospace industry, the scale of spacecraft required to meet the needs of space exploration is increasing. The overall mass increases and the stiffness decreases if the spacecraft is enlarged without optimizing the structure. Thin-membrane structures can reduce rockets’ launch mass and stowage volume due to the advantages of light mass, high stowage ratio, and flexible configuration [1]. Therefore, thin-membrane deployable structures are widely applied in thin-membrane antennas, [2] thin-membrane shields [3], and solar sails, [4,5] and play an essential role in the design of future large spacecraft. However, thin membranes are more sensitive to forces. The strain increases with increased membrane tension, which causes wrinkles. According to the present research, the two main states, pre-wrinkling and post-wrinkling, are attained depending on whether the strain of the membrane is considered or not [6]. When the strain of the membrane is not considered, the membrane is in the pre-wrinkling state; when the strain of the membrane is considered, the membrane is in the post-wrinkling state.

Currently, there are substantial studies on the post-wrinkling membrane, which mainly focus, on the following three theories: tension field theory, the Föppl–von Kármán theory, and the Koiter theory. Tension field theory was proposed to analyze the wrinkling behavior of membranes under finite deformation and is used to solve the problems that arise during the stretching of membranes [7,8,9,10,11,12,13]. Pipkin [9] applied the tension field theory by introducing the lowest possible energy density method matching a given deformation gradient. Steigmann [10] derived a partial differential equation for the membrane tension trajectory to avoid the membrane deformation in uncertainty. The Kármán theory is commonly used to analyze the amplitude and wavelength of thin membranes [14,15,16,17,18,19,20,21,22], and it takes into account the effects of membrane bending in the analysis of wrinkles. Fu et al. [21] established a solution framework for compressible and incompressible materials under the Föppl–von Kármán Theory. Huang et al. [22] used the Föppl–von Kármán theory to improve the accuracy and efficiency of membrane instability calculations by transitioning the Fourier-approximated membrane model to a multiscale model through the Arlequin method. Koiter’s theory is based on the three-dimensional elastic theory, which simulates the fold mode by combining the stretching and bending effects of the thin plate [23,24]. Taylor et al. [25] applied Koiter’s nonlinear plate theory to the numerical simulation of stretching elastic thin plate wrinkles. Ciarlet [26] proposed a natural two-dimensional Koiter model for linear elastic shells under half-space constraints. In addition to the studies above, some scholars have combined the calculation of post-wrinkling membranes with finite element theory. Lee et al. [27] used the nonlinear finite element method to simulate the large deformation of membrane wrinkles in migration coordinates. Wong et al. [28] analyzed the generation and increase of wrinkles in corner-point tensioned square membranes by finite element simulation software. Wong’s method could accurately predict the central membrane wrinkles, but not the wrinkles at the edges.

Compared with the post-wrinkling theories of thin membranes, the pre-wrinkling theories are less well studied. Silvestre [29] applied the modal method to unidirectional tensioned thin plates and proposed analytical solutions for nonuniform displacement, stress, and strain fields, which laid the foundation for the future mechanical roots of membrane wrinkle generation. Martins et al. [30] performed a pre-wrinkling analysis based on the modal idea of generalized beam theory using the principle of potential energy standard value. Li et al. [31] proposed a stress superposition method (SSM) and predicted the location of wrinkles in rectangular membranes with different aspect ratios. However, a stress calculation model for regular polygons was not established in their studies. Due to the complexity and inhomogeneity of the membrane stress field, there are relatively few solutions for stresses in orthogonal polygonal membranes determined by analytical methods. This study will address the above problems.

The purpose of this paper is to establish a pre-wrinkle model for corner-point tensioned orthogonal polygonal membranes and put forward the idea of solving the stresses in orthogonal polygonal membranes by the SSM. The stress regions of the membranes with odd and even edges are divided. The expressions of the constant coefficients of the stresses in the orthogonal polygonal membranes are derived, and the analytical equations of the stresses in the orthogonal polygonal membranes are solved. The SSM and finite element simulation are used to solve different types of membrane samples, and the SSM is verified by comparing the analytical results with the simulation. Considering the sensitivity of membrane curved edge to the change in membrane stress, the response of the membrane curved edge arch height to the change in membrane mechanical properties is investigated. As shown by the analysis results, the SSM can be used to solve for the optimal height of the membrane arc edge and to predict the effect of the arc edge on wrinkle generation.

## 2. Stress Superposition of Edge-Shaped Membranes

When an object is subjected to an external force, it produces a small deformation. According to the article [31], a two-ended rope subject to tension F is equivalent to the superposition of two ropes, each of which has one end under tension and the other end fixed.

For flat (planar) membranes, the form of force changes. When tension is applied to each corner point of the membrane, the membrane plane has a two-dimensional force. Opposite to each tensioning point, there are multiple tensioning points that bear the tension. Because of the symmetry of the tension point, the rectangular membrane resembles the stress superposition of one-dimensional cable tension. When the displacement produced by the membrane under tension is small, it can be regarded as bearing a tension between two opposite points. This method is unable to solve regular polygonal membranes with odd tension points, such as regular triangles and regular pentagons. The stress superposition model of a regular polygon is explored through the force analysis of regular triangular membranes. As is shown in Figure 1, the three corner points of the triangular membrane are sorted clockwise as corner point 1, 2, and 3, and subjected to tensile force F. Based on the force analysis of an arbitrary point M, the force from corner point 1 to M is $F_{1M}$, that from corner point 2 to M is $F_{2M}$, and that from corner point 3 to M is $F_{3M}$.The following equations can be obtained according to the force balance.
$${\vec{F}}_{2M}+{\vec{F}}_{3M}={\vec{F}}_{1M}\ldots {\vec{F}}_{1M}+{\vec{F}}_{2M}={\vec{F}}_{3M}\ldots {\vec{F}}_{1M}+{\vec{F}}_{3M}={\vec{F}}_{2M}$$

When calculating the stress at point M, if the stress of the three tension F is superimposed on the M point, the stress is increased by two times. Therefore, if the size of the arbitrary point of the regular triangle does not change after the stress superposition, it is equivalent to the superposition of three membranes with two points fixed and one point tensioned by F/2. The stress superposition form is shown in Figure 2.

Given the above theoretical analysis, the thin membrane stress of any regular polygon can be solved by a similar method. If the number of sides of a regular polygon is n, it can be transformed into a superposition of n polygonal stresses. Among them, each polygon has (*n* − 1) corner points fixed and one corner point tensioned by F/2. The direction of the tension is in the connection line from the center of the shape to the corner point. This method decouples the stress distribution of the regular polygon structure in the multi-point tension state into the superposition of the single-point tension state, so that the analytical solution of the stress field of the regular polygon membrane under the corner tension can be obtained. When using the stress superposition method to solve the stress distribution in this article, the membrane needs to meet the following conditions:(1)The membrane structure is a symmetrical regular polygon shape;(2)The membrane is subjected to symmetrical tensile forces, and the tensile forces are equal in magnitude;(3)The corner pull direction is on the line connecting the center of the regular polygonal membrane to the corner points.

## 3. Calculation and Verification of Positive Polygon SSM

### 3.1. Theoretical Derivation of SSM in Regular Polygons

According to the equation of Timoshenko and Goodier [32] for the corner point tension, the stress associated with the corner point *i* (*i* = 1,2,3,4) in the polar coordinate system can be described as:(1)$$\sigma_{r}=\frac{kT}{2r_{i}}\cos\alpha$$
where α is the angle between the line from point *i* to point M and the line where the corner point pull force is located; $r_{i}$ is the distance from point *i* to point M; $\sigma_{r}$ is the radial positive stress component; and *k* is the stress-solving coefficient. When the thickness of the membrane is *t*, the formula can be described as:(2)$$\sigma_{r}=\frac{kT}{2tr_{i}}\cos\alpha$$

According to the distribution law of the stress circle of the tension membrane at the corner, the tangency of the stress circle and the arc edge of the membrane is defined as the critical condition. The stress circle critical conditions of the odd-sided positive polygon and the even-sided positive polygon are different. The SSM of the odd-sided and even-sided membranes needs to be analyzed separately. In order to observe the law of stress superposition of regular n-polygons, regular hexagons and regular polygons are selected to calculate the stress superposition. Since the arc edge can improve the membrane’s mechanical properties, the arc edge of the membrane is taken into account in the stress superposition calculation of the regular polygon membrane. The stress region division is done for the ortho-hexagon, as is shown in Figure 3. For the regular hexagon, the stress arc distribution exists mainly in the following three regions: (I) the stress circle tangent to $\overset{⏜}{3-2}$ and $\overset{⏜}{6-5}$, which is a continuous arc in the membrane plane and not divided; (II) the stress circle tangent to $\overset{⏜}{4-3}$ and $\overset{⏜}{5-4}$, where the stress arc is divided into three segments; and (III) the stress circle crossing $\overset{⏜}{4-3}$ and $\overset{⏜}{5-4}$, where the stress arc is cut into five segments. With a gradual increase in the radius of the stress circle, each tangency is a critical point, and each critical point passing will be divided into two more stress arcs.

According to the balance between the stress on the membrane stress arc and the tension at the corner point, the following equations can be obtained.
(3)2∫0γ1kT2rcosθ·rcosθdθ=T2, a∈ℕI,
(4)2∫0γ2kT2rcosθ·rcosθdθ+2∫γ2+γ3γ1kT2rcosθ·rcosθdθ=T2, a∈ℕII,
(5)2∫0γ2kT2rcosθ·rcosθdθ+2∫γ2+γ3γ2+γ3+γ4kT2rcosθ·rcosθdθ+2∫γ2+γ3+γ4+γ5γ1kT2rcosθ·rcosθdθ=T2, a∈ℕIII
where, ℕI, ℕII, and ℕIII represent stress regions I, II, and III, respectively, and a is the location of the stress calculation point.

Then, *k* in the above equations is obtained as follows:(6)k1=2sin2γ1+2γ1−1, a∈ℕI,
(7)k2=2sin2γ2+sin2γ1−sin2γ2+γ3+2γ1−2γ3−1, a∈ℕII,
(8)k3=2sin2γ2+sin2γ1+sin2γ2+γ3+γ4−sin2γ2+γ3−sin2γ2+γ3+γ4+γ5−2γ3+2γ1−2γ5−1, a∈ℕIII

According to the relationship above, if the number of sides of the regular polygon is 2 m, when *m* > 2, the general expression of *k* can be obtained as follows:(9)$$k_{i}=2{\left(\begin{array}{l} \sin2\gamma_{2}+\sin2\gamma_{1}+\sin2\left(\gamma_{2}+\gamma_{3}+\gamma_{4}\right)-\sin2\left(\gamma_{2}+\gamma_{3}\right)\\ +\cdots +\sin2\left(\gamma_{2}+\gamma_{3}+\cdots +\gamma_{2i-2}\right)-\sin2\left(\gamma_{2}+\gamma_{3}+\cdots \gamma_{2i-3}\right)\\ -\sin2\left(\gamma_{2}+\gamma_{3}+\gamma_{4}+\cdots +\gamma_{2i-1}\right)+2\gamma_{1}-2\gamma_{3}-\cdots -2\gamma_{2i-1} \end{array}\right)}^{-1}i=2,3\cdots m$$

Above are the expressions for the stress circle distribution and the coefficients of radial stress $\sigma_{r}$ for a membrane with an even number of sides of a regular polygon. Next, the distribution of the stress circle of the membrane with an odd number of sides will be analyzed. Taking a regular heptagon as an example, the stress circle is divided as shown in Figure 4.

The regular heptagon has one more stress region than the regular hexagon. The first three stress regions are consistent with the regular hexagon, so it is only necessary to calculate the coefficient of the fourth region.
(10)2∫γ2γ2+γ3kT2rcosθ·rcosθdθ+2∫γ2+γ3+γ4γ2+γ3+γ4+γ5kT2rcosθ·rcosθdθ+2∫γ2+γ3+γ4+γ5+γ6γ1kT2rcosθ·rcosθdθ=T2, a∈ℕIV

The solution gives:(11)k4=2sin2γ2+γ3−sin2γ2+sin2γ2+γ3+γ4+γ5−sin2γ2+γ3+γ4+sin2γ1−sin2γ2+γ3+γ4+γ5+γ6+2γ1−2γ2−2γ4−2γ6−1, a∈ℕIV

It follows that when the number of sides of a positive polygon membrane is 2 *m* + 1, there is a total of m + 1 values of *k*. The first m coefficient, *k*, is the same as those of a positive 2 m-sided shape, and the m + 1 th coefficient is:(12)$$k_{i+1}=2{\left(\begin{array}{l} \sin2\left(\gamma_{2}+\gamma_{3}\right)-\sin2\gamma_{2}+\cdots +\sin2\left(\gamma_{2}+\gamma_{3}+\cdots +\gamma_{2i-1}\right)\\ -\sin2\left(\gamma_{2}+\cdots +\gamma_{2i-2}\right)+\sin2\gamma_{1}-\sin2\left(\gamma_{2}+\gamma_{3}+\cdots +\gamma_{2i}\right)\\ +2\gamma_{1}-2\gamma_{2}-2\gamma_{4}-\cdots -2\gamma_{2i} \end{array}\right)}^{-1},\;i=m$$
when a point in the membrane plane is given, the stress circle area can be determined according to different tension points. The corresponding angle $\gamma_{i}$ and distance $r_{i}$ from the point to the tension point can be found according to the geometric relationship. According to Equations (6), (7), (9) and (12), the corresponding *k* value is obtained. By bringing the above values into Equation (1) respectively, the radial stress value produced by each tensional point can be obtained. Then, from Equation (13), the radial stresses generated at each point are transformed from the polar coordinate system to the Cartesian coordinate system. The stresses generated at each point can be superimposed in the x and y directions, respectively, according to the SSM.
(13)$$\left\{\begin{array}{c} \sigma_{x}=\sigma_{r}\cos^{2}\eta \\ \sigma_{y}=\sigma_{r}\sin^{2}\eta \\ \sigma_{xy}=\sigma_{r}\sin\eta \cos\eta \end{array}\right.$$

In the equations above, η is the angle between the force direction of the tension point and the x-axis of the coordinate system The planar thin membrane’s first principal stress and the second principal stress can be obtained by bringing the superposition value in the Cartesian coordinate system into Equation (14). Then, the Von Mises stress at each point of the thin membrane can be obtained by bringing it into Equation (15).
(14)$$\left\{\begin{array}{c} \sigma_{1}=\frac{\sigma_{x}+\sigma_{y}}{2}+\sqrt{\frac{{\left(\sigma_{x}-\sigma_{y}\right)}^{2}}{4}+\sigma_{xy}{}^{2}}\\ \sigma_{2}=\frac{\sigma_{x}+\sigma_{y}}{2}-\sqrt{\frac{{\left(\sigma_{x}-\sigma_{y}\right)}^{2}}{4}+\sigma_{xy}{}^{2}} \end{array}\right.$$
(15)$$\bar{\sigma }=\sqrt{\frac{\sigma_{1}{}^{2}+\sigma_{2}{}^{2}+{\left(\sigma_{1}-\sigma_{2}\right)}^{2}}{2}}$$

### 3.2. Verification of SSM on Regular Polygons

To verify the accuracy of the SSM of the regular polygon, the Abaqus finite element analysis software can be used to establish the regular hexagonal and regular heptagonal membrane finite element models. It is verified by comparing the simulation value and the theoretical solution. The membrane material is polyimide. The radius of the inner tangent circle R of both membranes is 270 mm, and the membrane thickness is 0.05 mm. The four sampling points ($c_{1}$, $c_{2}$, $c_{3}$, $c_{4}$) are located on the line from the tension point to the center of the membrane. The distance between each two points is 60 mm, as shown in Figure 5. The distance from each tension point to the center of the membrane is 360 mm.

The membrane size and force form in Figure 5 is input to Abaqus for static analysis. The analysis unit is a three-node triangular membrane element in Abaqus (M3D3), and each corner bears 10 N of tension. To make the membrane in a static state, a fixed boundary can be applied to the center of the membrane. The finite element simulation results are shown in Figure 6. The values of the four sampling points are shown in Table 1.

Taking the stress value of point $c_{1}$ in a regular hexagonal membrane as an example, this point is in stress region I. When the corner point is subjected to a tensile force of 10N, its stress distribution is shown in Figure 7. The stress circle and arc $\overset{⏜}{5-6}$ intersect at point H, and the arch height of the arc side is h. The circumference angle corresponding to the arc $\overset{⏜}{1-H}$ is $\xi_{2}$, and the central angle corresponding to the arc $\overset{⏜}{6-H}$ is $\xi_{1}$. The central angle corresponding to the arc $\overset{⏜}{1-6}$ is ψ, the radius is R_1_, and the line $\bar{1-6}$ is L. The following assortment can be obtained:
(16)$$L=\frac{2\sqrt{3}}{3}\left(R+h\right)$$
(17)R1=h2+12L22h
(18)$$\xi_{2}=\arcsin\frac{\bar{1-H}}{2R_{1}}$$
(19)$$\psi =2\arcsin\frac{L}{2R_{1}}$$
(20)$$\xi_{1}=\frac{1}{2}\psi -\xi_{2}$$
(21)$$\gamma_{1}=\arccos\frac{\bar{1-H}}{L}$$
(22)$$\gamma_{1}+\xi_{1}=\frac{\pi }{3}$$

By solving Equations (16)–(22), $\gamma_{1}=50^{o}$ can be obtained, and by bringing this into Equation (6), k = 0.73 can also be obtained. The radial stress can be calculated as 203,490 pa by bringing k and L into Equation (1). Since the point is the center of the membrane, the radial stress of six corner points is equal. The radial stress generated at each angular point can be superimposed by transformation into the Cartesian coordinate system through Equation (13). Then, the equivalent stress at the point can be found as 610,470 pa by Equations (16) and (17). The angular relationship of each point in Figure 5 can be obtained through a geometric relationship, the corresponding k value and r value can be obtained, and then the equivalent stress at each point can be determined. The finite element simulation results and theoretical calculation results of each point are shown in Table 1.

From the results above, the average error of the n-hexagonal membrane is 3.15% and the average error of the n-heptagonal membrane is 4.25%. SSM can be used for both the regular hexagonal and regular heptagonal membranes to find their theoretical solutions under tension at the corner points. That is, regardless of whether the number of sides of the n-polygon is odd or even. Under the condition of being subjected to tension F at the corner point, this can be regarded as the stress superposition of regular polygonal membranes with n single points under tensile force F/2 and the remaining n−1 fixed points. It is worth noting that the mechanical boundary and membrane shape should be consistent with the model proposed in this paper during simulation and calculation. When the tensile force value, tensile force direction, and membrane shape change, this theoretical analysis method is no longer in use and needs to be analyzed separately.

## 4. Analysis of Mechanical Properties of Orthogonal Polygonal Membranes Based on SSM

According to the law of stress propagation in the membrane plane, the midpoint of the straight edge of the corner tensioned regular polygon membrane is the lowest point of the stress, which seriously affects the stiffness of the membrane plane. In previous studies of thin membranes, many scholars have demonstrated that the design of curved edges can effectively increase the stress level at the edges of the membrane and equalize the stress distribution of the membrane. The design of the curved edge should be such that the effective area of the membrane remains constant. Otherwise, the curved edge design will bring about a reduction in the working plane of the membrane. When the effective area of the membrane is constant according to Equation (15), the size of the membrane will increase with an increase in arch heigh. Therefore, a balance point should be found to exert reasonable control of the arch height of the curved edge. In this section, the finite element simulation is used to explore the stress distribution law of thin membranes. The optimal arc edge structure is obtained according to the SSM of regular polygonal thin membranes, which provides theoretical guidance for the future arc edge optimization design of thin membranes.

A finite element model is established by taking the straight line from the center point of the regular hexagonal membrane to the center point of the arc edge as the research object. When the arch height h is 0 mm, 10 mm, 20 mm, 30 mm, and 40 mm, the stress distribution of the membrane plane where the straight line is located is studied. The membrane material is polyimide, with a thickness of 0.05 mm and an effective circle radius r of 270 mm. The tensile force of 10 N is applied to each corner point and the analysis results are shown in Figure 8.

When the effective area of the membrane is constant, with the increase of arch height, the stress at the center of the film shows a downward trend, and the stress at the midpoint of the arc shows an upward trend. The increase of the stress at the midpoint of the arc edge is greater than the decrease of the stress at the center of the membrane. When the stress at the center point of the membrane is equal to the stress at the center point of the arc edge, the overall minimum stress of the membrane is enhanced to the greatest extent. At this time, the highest stiffness is obtained in the whole membrane plane. The trend of stress at the center point of the membrane and the trend of stress at the center point of the arc edge are obtained by the SSM, as shown in Figure 9.

When the two figures intersect at a single point, the stress at the center point of the membrane is equal to the stress at the center of the arc edge. The minimum stress in the whole membrane plane reaches the maximum value, in which the peripheral size of the ortho-polygon membrane only increases by 4.6%, but the minimum stress on the membrane surface is increased by 18.52%. For any orthogonal polygon structure, the law of stress propagation is the same, and the SSM can obtain the stresses at each point. According to the intersection point of the figure, the arch height corresponding to the maximum stiffness of the membrane is determined.

In addition, the design of the curved edges of the membrane changes the initial stress distribution state of the membrane, which can impact the wrinkling of the membrane. In previous studies, it was considered that the membrane would be wrinkled at the first principal stress $\sigma_{1}>0$ and the second principal stress $\sigma_{2}<0$ [6]. In the actual membrane tensioning, due to the membrane strain generation, wrinkles are easy to generate in the region where the second principal stress of the membrane is small, according to the tension field theory and relaxation strain theory. Therefore, it is necessary to investigate the effect on the region where membrane wrinkles may be generated when the stress at the midpoint of the arc is equal to the stress at the center of the membrane. In order to facilitate observation, the regular hexagonal membrane with arch heights of 0 mm, 10 mm, and 12.5 mm is analyzed by the finite element method. The second principal stress distribution is shown in Figure 10, and the second principal stress distribution on the straight line from the center of the membrane to the center of the arc edge is shown in Figure 11.

From the results of the analysis above, the second principal stress distribution states of the three are basically in the same state, and all of them are greater than 0. The second principal stress in the middle is higher than the second principal stress in the edge, which indicates that the wrinkles of the membrane can easily occur in the edge area. When the arch height of the arc side increases from 0 mm to 12.5 mm, the stress in the middle area of the membrane is unchanged, but its second principal stress increases by 143%. This shows that the design of the curved edge not only improves the stiffness of the membrane, but also reduces the possibility of wrinkling at the edge of the membrane.

## 5. Conclusions

In this paper, the SSM for regular polygonal membranes is proposed for the first time. Taking regular hexagonal and regular heptagonal membranes as research objects, the stress superposition algorithm for regular n-polygons membranes is derived. The accuracy of the SSM in the calculation of ortho-hexagonal membranes is verified by comparing the stresses at sampling points of ortho-hexagonal and ortho-heptagonal membranes with the results of finite element simulation. The curved edge of corner point tensioned regular polygonal membranes is more sensitive to stress. It is proposed for the first time to optimize the arc edge of the membrane in order to improve the maximum stiffness of the membrane while keeping the effective area of the membrane unchanged. Through the stress calculation of the thin membrane by the SSM, the optimal value of the arch height can be regarded as the height at which the stress at the middle point of the arc edge of the membrane is equal to the stress at the center of the membrane. The analysis of the diametrical principal stresses of the membrane predicts that the wrinkles of the regular polygon membrane may appear in the edge region of the membrane. The design of the curved edges can substantially increase the second principal stress at the edges of the membrane, thus reducing the possibility of wrinkles.

This analysis improves the theoretical basis of stress calculation for corner point tensioned regular n-polygons membranes. That is, for any corner point tensioned regular polygonal membrane structure, the stress field can be regarded as the superposition of the stress distribution field generated by half of each load. The proposal of this method will play an important role in the calculation of the stress field of regular polygonal membranes under corner tension, the design of the arc, and the prediction of membrane wrinkles in the future. It provides theoretical support for the development of regular polygonal membrane sunshields, membrane solar sails, and membrane solar arrays in spacecraft.

## Figures and Tables

**Figure 1 materials-15-00192-f001:**
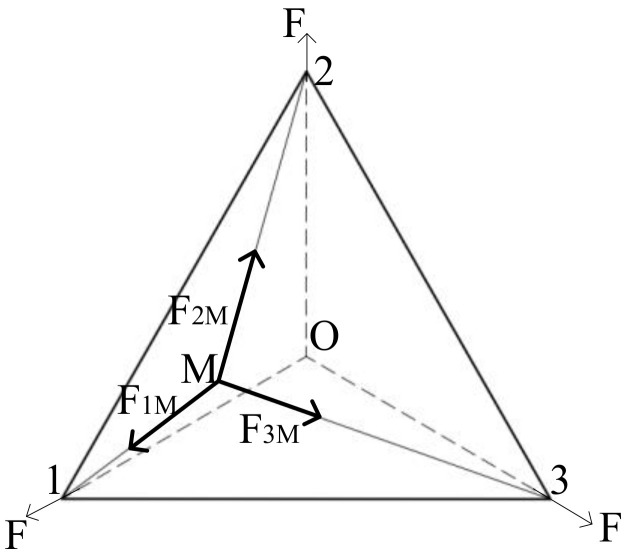
Analysis of the force at any point of the positive triangle.

**Figure 2 materials-15-00192-f002:**
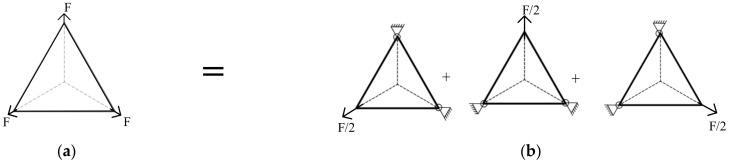
Triangular membrane stress superposition. (**a**) A plane membrane with three points subject to tension F; (**b**) Double point fixation, single point force F/2 of the plane membrane.

**Figure 3 materials-15-00192-f003:**
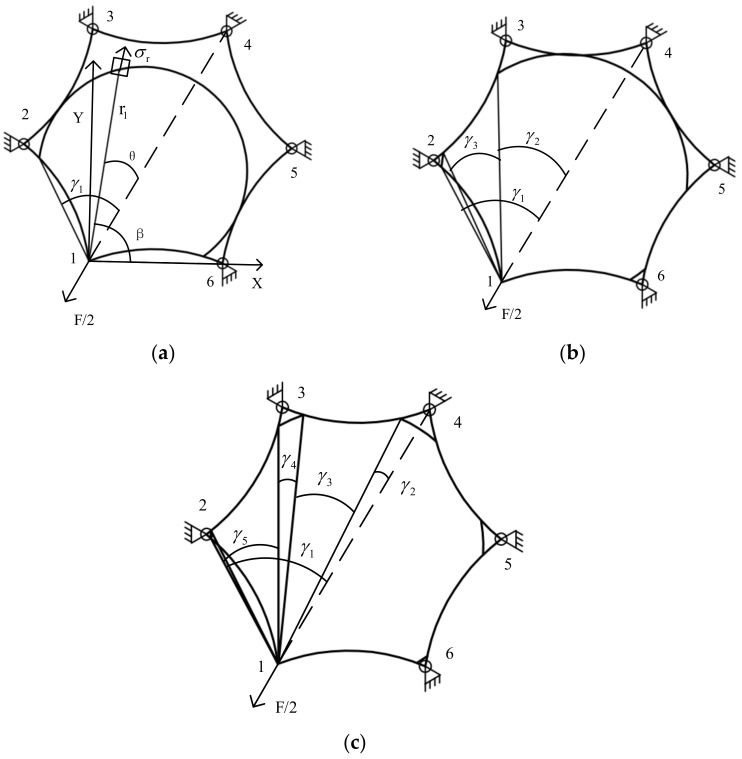
Corner point tensioning ortho-hexagonal stress area division. (**a**) Region I stress arc distribution;(**b**) region II stress arc distribution; (**c**) region III stress arc distribution.

**Figure 4 materials-15-00192-f004:**
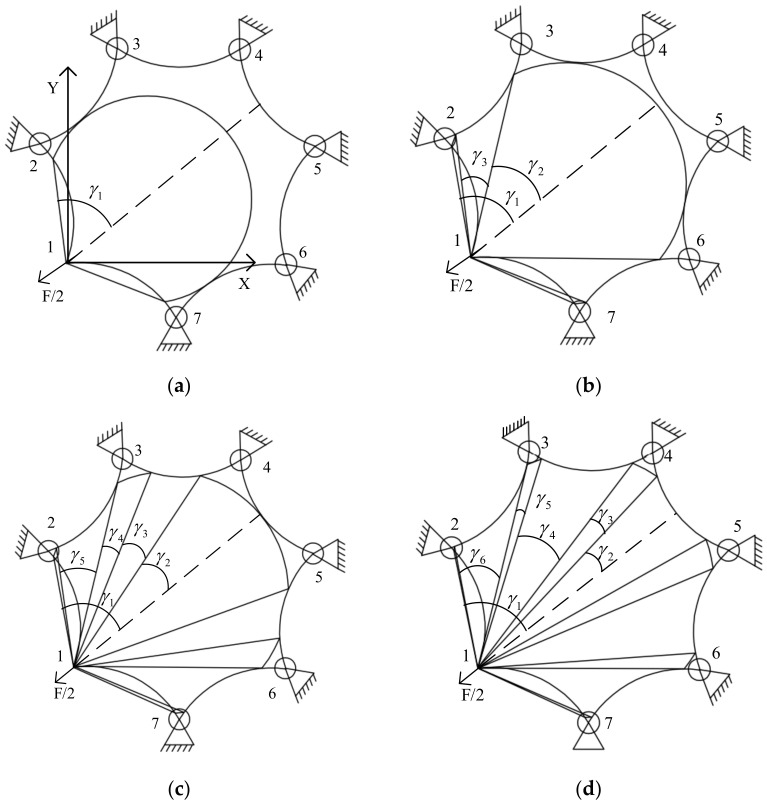
Corner point tensioning positive heptagonal stress area division. (**a**) Region I stress arc distribution; (**b**) region II stress arc distribution; (**c**) region III stress arc distribution; (**d**) region IV stress arc distribution.

**Figure 5 materials-15-00192-f005:**
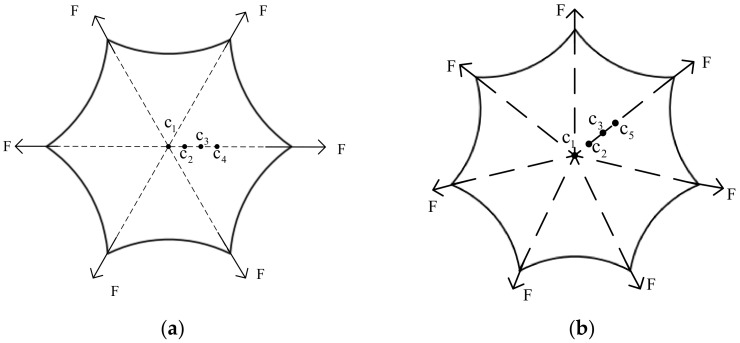
Example of stress superposition in ortho-hexagonal and ortho-heptagonal membranes. (**a**) Positive hexagonal sampling point distribution; (**b**) Positive heptagonal sampling point distribution.

**Figure 6 materials-15-00192-f006:**
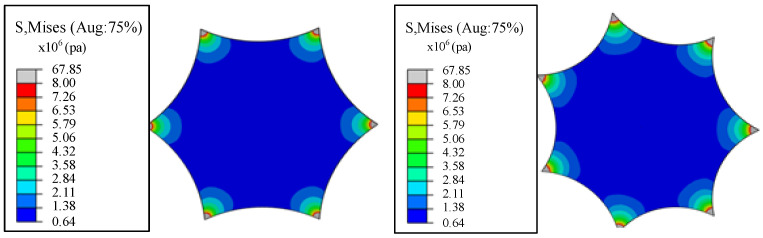
Finite element simulation stress distribution.

**Figure 7 materials-15-00192-f007:**
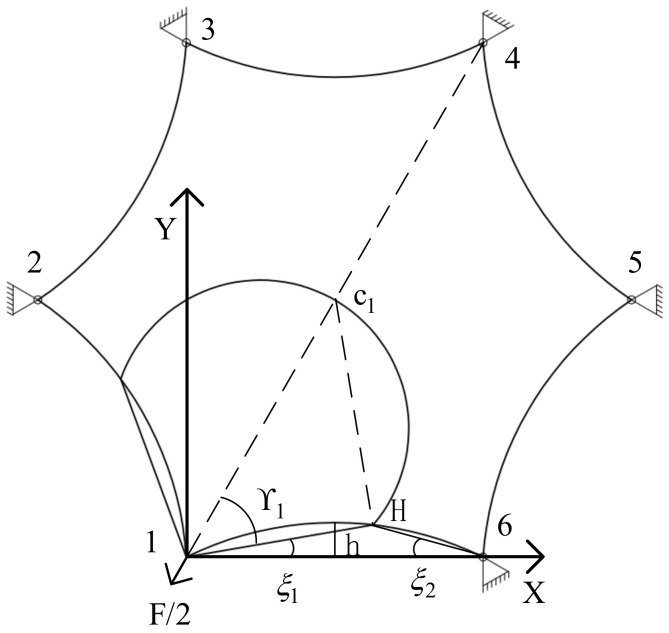
$c_{1}$ nodal stress calculation.

**Figure 8 materials-15-00192-f008:**
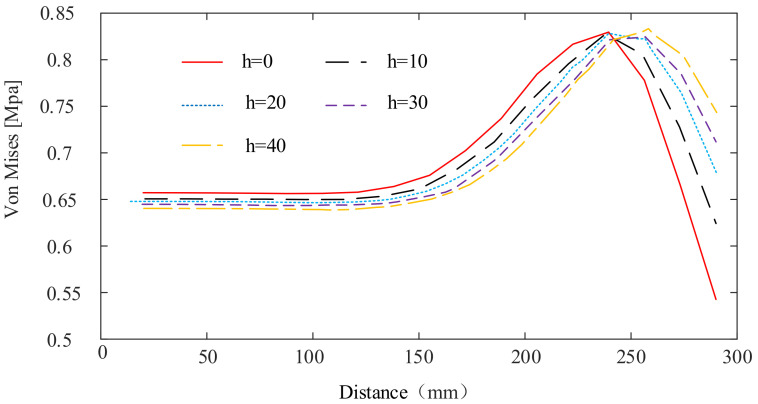
The stress distribution from the center point of the membrane to the midpoint of the membrane arc at different arch heights.

**Figure 9 materials-15-00192-f009:**
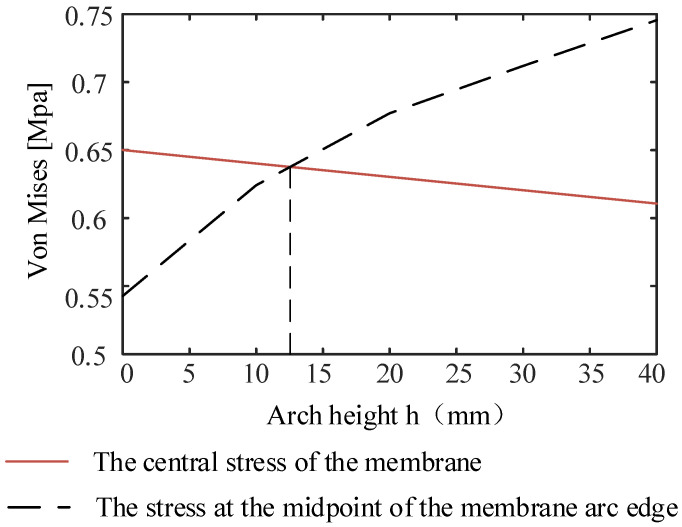
Stress variation in the center of the membrane and the center of the curved edge.

**Figure 10 materials-15-00192-f010:**
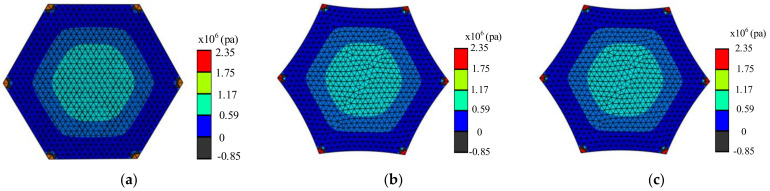
Distribution of the second principal stress in the membrane at different arch heights. (**a**)The arch height h = 0 mm; (**b**) the arch height h = 10 mm; (**c**) the arch height h = 0 mm.

**Figure 11 materials-15-00192-f011:**
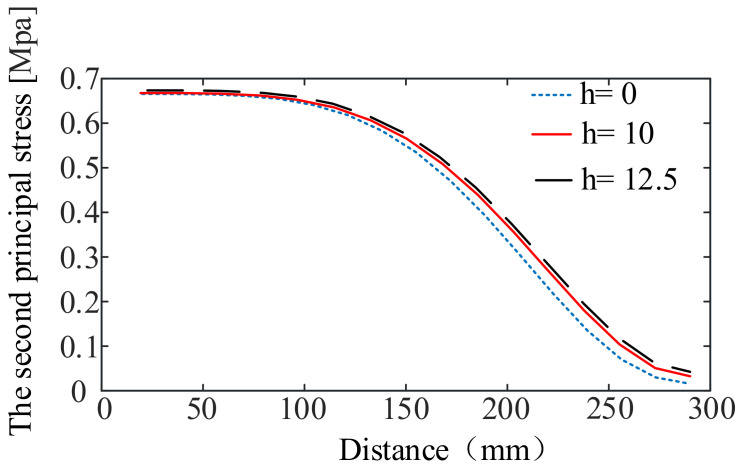
The second principal stress distribution on the straight line from the center of the membrane to the center of the arc edge.

**Table 1 materials-15-00192-t001:** Comparison of the simulation and theoretical values of both membranes.

Shape	Nodes	Simulation Value [Pa]	Theoretical Value [Pa]	Error
Regular hexagon	$c_{1}$	640,656	610,470	4.7%
	$c_{2}$	640,729	618,059	3.5%
	$c_{3}$	649,116	631,387	2.7%
	$c_{4}$	775,573	762,149	1.7%
Regular heptagon	$c_{1}$	746,736	698,081	6.5%
	$c_{2}$	756,771	702,967	7.1%
	$c_{3}$	759,641	747,279	1.6%
	$c_{4}$	830,996	816,430	1.8%

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
