# Peer review of "Stress Superposition Method and Mechanical Properties Analysis of Regular Polygon Membranes"

_materials, 2021, doi:10.3390/ma15010192_

Round 1

Reviewer 1 Report

The mistakes noticed in the text of the work are shown in the attachment

Reviewer 2 Report

Authors present an analytical procedure to predict or estimate the stress distribution in 2D polygonal membranes. There are several issues to address before it reaches a publishable form:

  1. Authors did not follow the journal guidelines; reference style is not the correct one.
  2. changes in font type through the manuscript are a clear evidence of a careless submission.
  3. In correct numbering of the sections.
  4. Details for the FEA simulations are missing, type of element, material used, dimensions of the samples, software used, etc.
  5. Authors should be clear and state the limitations of their approach. For example up to which degree of symmetry or rotational symmetry does the approach is valid?
  6. Is the approach considering stresses through the thickness? If not this is a plane stress case and it should be noted.
  7. Details on the boundary conditions (BCs) used both analytically and FEA should be specified.
  8. Also, discussion on how the model changes if this BCs are modified should be included.
  9. Some thoughts on applicability of the work should be included.
  10. Page 8 line 199, missing or wrong symbol.
  11. Change “Mises” to “von Mises”
  12. Figure 8 is odd, why h is label and axis at the same time?

Round 2

Reviewer 2 Report

I still do not see the type of element use in their FEA simulations.

Author Response

Dear review experts;

Regarding the unknown analysis unit in the article, I have modified it.

The analysis unit is a three-node triangular membrane element in Abaqus (M3D3).

Thank you very much for your comments.